# An Energy Efficient Enhanced Dual-Fuzzy Logic Routing Protocol for Monitoring Activities of the Elderly Using Body Sensor Networks

**Sea Young Park [1], Dai Yeol Yun [2], TaeHyeon Kim [2], Jong-Yong Lee [3] and Daesung Lee [4],***

[1]  Department of Immersive Content Convergence, KwangWoon University, Seoul 01897, Korea;
   android5966@kw.ac.kr
[2]  Department of Plasma Bioscience and Display, KwangWoon University, Seoul 01897, Korea;
   hibig10@kw.ac.kr (D.Y.Y.); surowang@kw.ac.kr (T.K.)
[3]  Ingenium College of Liberal Arts, KwangWoon University, Seoul 01897, Korea; jyonglee@kw.ac.kr
[4]  Department of Computer Engineering, Catholic University of Pusan, Busan 46252, Korea
*  Correspondence: dslee@cup.ac.kr; Tel.: +82-10-8733-9871

**Abstract:** Wireless body area networks (WBANs) are an important application in wireless sensor networks (WSNs). Specifically, in healthcare monitoring systems, it is important to screen the patient's biometric signals. For example, the elderlies' vital signs, such as ECG (Electrocardiogram), blood pressure, heart rate, and blood glucose, can be used as measures of their well-being and are all critically important for remote elderly care in tracking their physical and cognitive capabilities. Therefore, WBANs require higher energy efficiency and data transmission. This paper proposes a cluster-based routing protocol which is suitable for WBANs while analyzing energy efficiency issue in data transmission. Considering the importance of sensor nodes in a specific environment for improving the network's lifetime, the protocol based on the LEACH (low energy adaptive clustering hierarchy) algorithm is proposed. Due to its avoidance of long-distance transmission, the clustering technique is an efficient algorithm for prolonging the lifetimes of sensor networks. Therefore, this paper suggests an enhanced LEACH-dual fuzzy logic (ELEACH-DFL) protocol based-on clustering for CH (cluster head) selection and cluster configuration in wireless sensor networks. The simulation and analysis results address that the enhanced algorithm reduces the energy consumption effectively and extends the lifespan of the entire network. For wired sensors, attaching sensors to the user may cause problems and inconvenience of mobility. This leads to the use of wireless sensors to proceed with body sensors, which should consider the problem of battery efficiency, which concerns the configuration of wireless sensors. The LEACH protocol is energy efficient until the first node dead is generated. However, there is a sharp drop in energy efficiency after that. The ELEACH-DFL protocol has the advantage of maintaining energy efficiency even after the first node dead is generated, with the utmost consideration being given to stability in consideration of cluster selection and cluster head selection. In a field of $50 \times 50$, the FND efficiency improvement rate of ELEACH-DFL versus LEACH protocol is approximately 32%. In addition, in a field of $50 \times 150$, the FND efficiency improvement rate of ELEACH-DFL versus LEACH protocol is approximately 159%.

**Keywords:** WBAN; healthcare; routing protocol; energy efficiency; LEACH; fuzzy logic

---

## 1. Introduction

With the development of industry, the aging of mankind, and increasing medical costs, research on the new healthcare system has expanded. As semiconductor manufacturing technology advances, tiny implanted (Bio) sensors are used to measure the biometric signs. These sensors can analyze and

store measured data, which can also be transmitted to an external device such as a medical server to diagnose the patient's or the elderly's status. For that purpose, hardwired connections are difficult for patients to wear, and expensive for deployment and maintenance. This makes it easier and cheaper to apply sensors to patients using a wireless interface [1,2].

The increasing usage of wireless network and the miniaturization of sensors have bolstered the development of wireless body area networks (WBAN). In these networks, various sensors are attached to clothes or bodies or implanted under the skin. The wireless characteristics of the network and various sensors provide numerous new, practical, and innovative applications to improve healthcare and the quality of life. Body sensor networks can help people live comfortably by providing users with various activities and behavioral monitoring servicers through applications in healthcare, emergency treatment, fitness, etc. [3].

Users can screen their health status through various sensors in real time with personal desktops and smartphones. Medical institutions can obtain their sensor information and analyze it remotely or offline. By collecting and analyzing users' biometric information from sensors, body sensor-based medical services can provide users with accurate medical services. Due to the convenience of mobile devices such as smartphones and the development of WBAN, research on mobile healthcare is actively carrying on [4].

Mobile healthcare is expected to facilitate the prevention and management of diseases, since the sensor information can be obtained through mobile devices and the medical service area can be expanded to the observer rather than the medical institution center. Therefore, mobile healthcare systems can provide the users with ease of use, reduced risk of infection, reduced risk of failure, reduced user discomfort, and lower cost of care delivery [5,6].

The functionalities of a mobile healthcare system are:

1. To alert its user of the approach or development of a potential medical emergency, so that precautionary action can be taken.
2. To alert the medical emergency system if vital signs drop below a certain threshold.
3. To measure a real-time bio-signal for local processing.

WBANs imply a ubiquitous environment in which sensor devices have formed a network near the human body. Unlike sensor networks, sensor devices deployed in WBANs are very small in size, and there is another limitation that requires a long operational lifetime of the sensors: it is difficult to replace or recharge batteries in cases in which sensors are placed in a person's body or clothing. Therefore, a sensor device's energy technology is a very important in WBAN.

One of innovations used in WBAN to extend the lifetimes of sensors is to use transmission power control algorithms. The transmission power control algorithm regulates the transmission power of a transmitter to reduce energy consumption for every transmitting channel [7]. Existing transmission power control algorithms perform transmission power regulation based on the closed-loop mechanism. A closed-loop-based algorithm means a method in which a node sensor, usually called cluster head or sink, informs other sensors of transmission power level through a control message channel when conducting communication between a sensor and a cluster head (CH). A sensor can transmit the collected data with the transmission power received from CH. This closed loop mechanism, however, has the disadvantage of high energy consumption due to excessive control messages [8].

Another innovation used in WBAN to extend the lifetimes of sensors is a cluster-based routing protocol. Due to various restrictions on wireless networks, routing protocols used in networks are also subject to a number of constraints. Many studies have been conducted in this field and two types of topological structures have been proposed, primarily planar topologies and layered topologies. In a planar structure, all nodes in a network are at the same level and have the same routing capabilities, making it simple and efficient in a small network. The problem, however, is that as the network grows, the amount of routing information increases rapidly, and it takes a long time for routing information to reach the final node. For large networks, cluster-based hierarchical routing can be used

to resolve the problem. In hierarchical routing, nodes within the network are dynamically configured by being grouped into areas called clusters, and clusters are eventually assembled into base stations. Routing with clustering has the following advantages:

1.  Clusters help maintain a relatively stable network topology.
2.  Routing overhead can be significantly reduced by propagating high levels of information through cluster heads.
3.  Only CH or intermediate nodes need to maintain path information.
4.  Reduce energy consumption across all networks.
5.  Improve network scalability.

Consequently, the use of hierarchical routing protocols will maximize the energy efficiency. Optimal CH selection and clustering configuration methods are required to ensure equal energy consumption to maximize the network lifetime in a routing protocol. In this paper, we propose a hierarchical routing protocol and utilize fuzzy logic in the method of optimal CH selection and clustering configuration.

The rest of this paper is as follows. In Section 2, we introduce the related research. More details of enhanced LEACH-dual fuzzy logic (ELEACH-DFL) with optimal CH selection method and clustering configuration are presented in Section 3. In Section 4, we evaluate the performances of the ELEACH-DFL. Finally, the paper is concluded in Section 5.

## 2. Related Researches

### 2.1. Body Sensor Network

WBAN is one of the wireless sensor network technologies introduced to monitor health conditions by attaching a tiny biosensor to the body so as not to interfere with normal human activity. A body sensor is a piece of technology that can detect body temperature, blood pressure, or one of various other external stimuli. The collected sensor information is wirelessly transmitted to the external device and messaged to the caregiver who is responsible for the patient, in real time. In the event of an emergency or hazardous situation, the system can send the collected sensor information in the form of messages and alarms to caregivers. As such, WBAN aims to improve human quality of life by providing real-time support for practical applications at low cost [9].

Figure 1 shows the basic framework of healthcare for the elderly in WBSN. The architecture consists of several sensor nodes and a coordinator node. The coordinator manages, collects, stores, and analyzes data received from the sensor nodes to connect to the WBSN for adjustment and monitoring. Wearable sensor nodes measure and process biophysical parameters, such as heart rate, body temperature, and blood pressure to send data to the fog server or base station (BS). On the sensor node side, the software architecture defines and extends a flexible modular architecture, making it important to provide common signal processing capabilities that are readily available [10].

To develop and deploy the WBSNs, below features should considered together, but it is hard to satisfy all these requirements. Therefore, a system that meets several requirements should be established depending on the specific application.

1.  Data quality: The quality of data is of a high standard to ensure that the decisions made are based on the best information possible.
2.  Data management: The need to manage bio-datasets is of utmost importance.
3.  Sensor validation: It is of the utmost importance, especially within a healthcare domain, that all sensor readings are validated. This helps to reduce false alarm generation and to identify possible weaknesses within the hardware and software design.
4.  Data consistency: Critical patient data may be fragmented across multiple networked PCs or laptops within the WBAN across multiple nodes.

5.  Security: WBAN transmissions require considerable effort to be secure and accurate.
6.  Interoperability: WBAN systems must ensure seamless data transfer between standards such as Bluetooth and ZigBee to facilitate information exchange and plug-and-play device interaction.
7.  System devices: The sensors used in the WBAN must be low in complexity, small in form factor, lightweight, energy efficient, user-friendly, and reconfigurable.
8.  Energy vs. accuracy: The sensor start-up policy should be determined to optimize the trade-off between the BAN power consumption and the probability of misclassification of patient health. High power consumption often results in more accurate observations of the patient's health status and vice versa.
9.  Privacy infringement: If WBAN technology exceeds "security" medical uses, people may think it is a potential threat to human freedom. Social acceptance of this technology will be the key to finding it in a wide range of applications.
10. Interference: The wireless link used for body sensors should reduce the interference and increase the coexistence of sensor node devices with other network devices available in the environment.

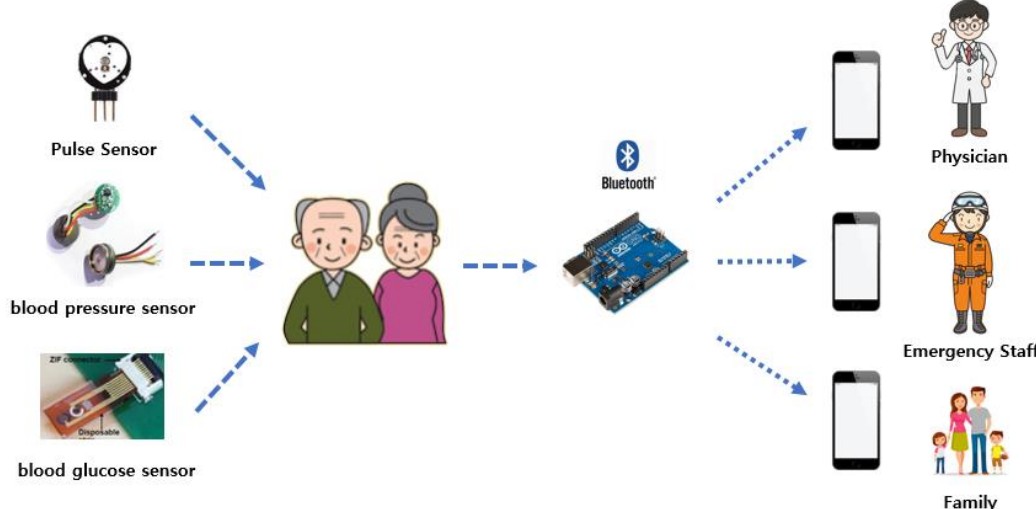

**Figure 1.** Framework for healthcare for the elderly using wireless body area networks (WBANs).

Here is some related research regarding the MAC layer. One of main goals of MAC protocol is to reduce power consumption from sources such as idle times, bandwidth, overhearing, and collision. The composition of the BSN can be expressed as shown in Figure 2.

Omeni et al. proposed an MAC protocol for a star-networked WBAN that supports TDMA to reduce the probability of collision and idle listening. Each slave node is assigned a slot by a central node. In any node, when an alarm is generated, additional slots for direct communication with the node can be allocated. This network connection system greatly reduces the possibility of collisions and idle listening, and can save significant amounts of power. In addition, time-slot allocation is dynamically controlled by the master, so a slave time slot could be changed every time by communicating with the master. This enables the system to better cope with fluctuating traffic [11].

An H-MAC protocol uses rhythm information of the heartbeat of a human being to perform time synchronization of a TDMA. Accordingly, the biosensor can achieve time synchronization without turning on the radio. The algorithm is verified with actual data but assumes a specific buffer. The simulation does not show any energy gain, and the protocol is designed only for the star-topology WBAN. When considering wireless transmission around and on the body, important issues are radiation absorption and heating effects on the human body. To reduce tissue heating, the radio's transmission power can be limited, or traffic control algorithms can be used [12].

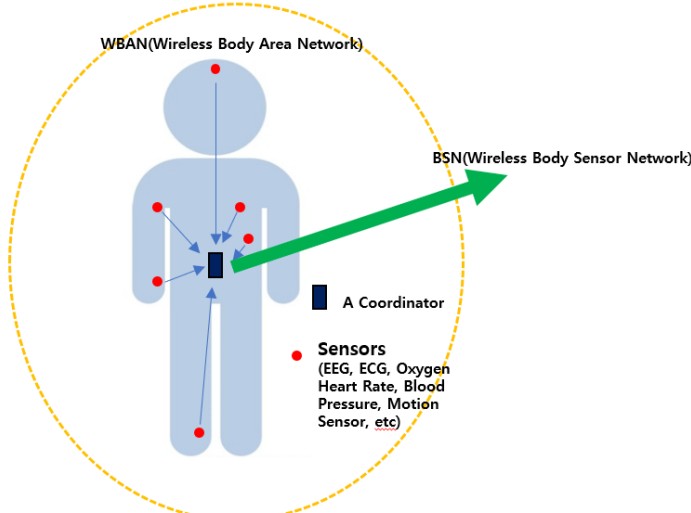

**Figure 2.** Example of patient monitoring in a wireless body area network.

Rate control is used to reduce bioeffect in a single hop network. Another possibility is a protocol for balancing the communication of the sensor node. The bioeffects model is both near-field and far-field, in relation to specific absorption rate (SAR). A normalized bioeffect metric, equivalent coefficient of absorption and bioeffects (CAB), was derived to evaluate and design the communication protocols for wireless biosensor networks. This shows that the bioeffects can be reduced via power scheduling and traffic control algorithms [13].

The thermal aware routing algorithm (TARA) model routes data away from high temperature areas (hot spots). Some factors lead to temperature increase, and the process for calculating SAR and the temperature increases of implanted biosensors uses the finite-difference time-domain (FDTD) method. The CH is switched based on the leadership history and the sensor locations. Packets are withdrawn from heated zones and rerouted through alternate paths. TARA suffers from low network lifetime and a high ratio of dropped packets, and does not take reliability into account [14].

The improvements to TARA are LTR (lowest routing) and ALTR (adaptive least temperature routing), which reduce unnecessary hop loops by maintaining a list of packets on the node most recently accessed. An ALTR converts to shortest hop routing when it reaches a predetermined number of hops to reduce energy consumption and heat generation on a network. This algorithm routes the packet to the coolest adjacent node without inducing a routing loop. In comparison with the shortest hop routing algorithm, this algorithm is much better executed in terms of reducing heat, delay, and power consumption generated [15,16].

"AnyBody" is a data gathering protocol that uses clustering to reduce the number of direct transmissions to the BS. Each node sends out a hello message in which it puts its unique identifier. No message is sent if the node has a higher density than all its neighbours. The join messages are relayed by the receiving nodes until it reaches a node with highest local density. Through this messaging process, the CH is elected to be the node with the highest density and nodes are grouped into clusters. After clustering, nodes need to be interconnected. All these connections beteen CHs will form a virtual backbone network. Finally, the routing paths are set up. A network in this protocol has a constant number of clusters, even though the number of sensor nodes is increased [17].

Another improvement of LEACH is hybrid indirect transmissions (HIT) which uses clustering to reduce the number of direct transmissions to the BS and allows parallel and multi-hop indirect transmissions, even in the case of multiple and adjacent clusters. The energy efficiency is improved but data reliability is not considered [18].

CICADA (cascading information retrieval by controlling access with distributed slot assignment) uses a data gathering tree and controls the communication using distributed slot assignment [19,20].

This protocol supports multi-hop routing and improves WASP (wireless autonomous spanning tree protocol). Each transmission cycle, the tree structure is used to allocate transmitting time slots to the different nodes in a distributed manner. CICADA divides such a cycle in a control sub-cycle and data sub-cycle, thereby lowering the delay and introducing mobility robustness. Data aggregation and the use of a duty transmission cycle improved the lifetime of a network [21–27].

*2.2. LEACH Protocol*

LEACH (low-energy adaptive clustering hierarchy) is a representative protocol based on cluster-based routing. The LEACH protocol consists of a set-up phase and a steady-state phase. In the setup phase, CHs are selected randomly by the stochastic threshold, and then clusters are configured. CHs from the nodes are selected by Equation (1) and the values are between 0 and 1 [28–31].

$$T(n) = \begin{cases} \dfrac{p}{1-p\left(r \bmod \frac{1}{p}\right)} & if\ n \in G \\ 0 & otherwise \end{cases} \tag{1}$$

where $p$ is the selection probability of the cluster head, $r$ is the current round, and $G$ is the set of nodes that were not elected as the cluster head until the previous round. The value of Equation (1) shall be between 0 and 1. Each node generates a random number value between 0 and 1 and is compared to Equation (1); then a node with a random number value less than the result of Equation (1) is elected as the CH. After CHs are selected, clusters are formed as follows. The selected CHs broadcast advertising messages, including their own data, to surrounding nodes. Nodes that receive advertising messages from CHs form a cluster by sending a join-request message to join the CH which is transmitting with the highest signal power; nodes can read the RSSI (received signal strength index). When all clusters' configurations are completed, CH creates the time division multiple access (TDMA) schedule that tells each node the time to transmit, depending on the number of member nodes.

At the steady-state stage, nodes in the cluster transmit data according to the TDMA schedule assigned by its CH selected during the setup phase and return to the sleep mode to save the power. After all nodes send data to the CH, the CH completes the steady-state phase by merging the data received from all nodes to the BS in a CDMA (code division multiple access) manner. The cycle that has been completed from these setup phase to the steady-state phase is called one round.

Figure 3 shows one round of flow-chart of LEACH; the left part is for CH selection, and the right part for cluster configuration. The dotted line is the radio communications of CH's broadcasting, TDMA scheduling, and nodes' messages.

The LEACH protocol has the advantage of increasing network energy efficiency by electing all nodes as cluster heads evenly, improving the problem of a certain node becoming a cluster head continuously in an existing clustering-based protocol, and distributing equally the energy consumption of the network. However, by using only an equation probabilistically when selecting a CH, the network has the disadvantages of shortening the lifespan of the network due to problems such as the selection of nodes with insufficient remaining energy. To address this, a variety of protocols have been proposed that modify the equation.

In the case of Equation (1) used to select CHs in the LEACH, the residual energy of the nodes is not taken into account. Therefore, even if the remaining energy of the actual node is low, it can be elected as a CH. To improve this, M.J. Handy proposed a threshold Equation (2) which was modified as follows.

$$T(n) = T(n) \times \frac{E_{current}}{E_{max}} \tag{2}$$

In Equation (2), $E_{max}$ is the initial energy as the maximum energy of the node and $E_{max}$ means the remaining energy of the node at specific round. The threshold equation's value is multiplied by the percentage of remaining energy in each node. This lowers the probability of being selected as the CH, as shown in Table 1.

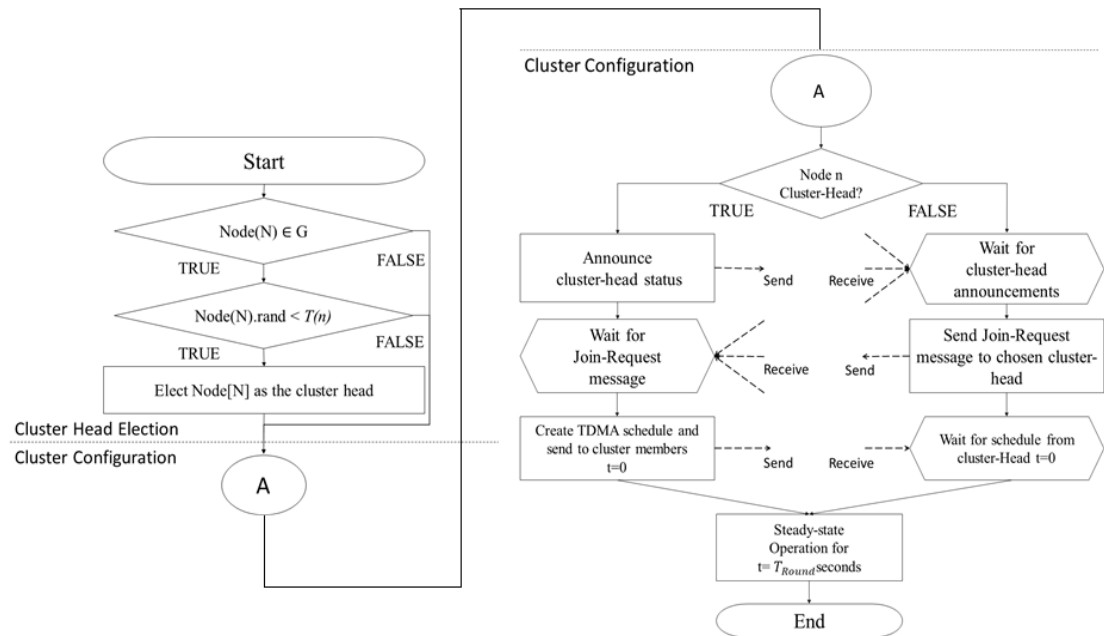

**Figure 3.** One round of flow-chart of LEACH.

**Table 1.** Change in the selection probability for being CH due to residual energy.

| $T(n)$ | $\frac{E_{current}}{E_{max}}$ | $T(n)_{new}$ |
|---|---|---|
| | 1 | 0.7888 |
| 0.7888 | 0.5 | 0.3944 |
| | 0 | 0 |

To make the nodes which have less remaining energy not induce CHs results in increased network lifetime. In addition, a number of variations of LEACH have been studied to improve the performances of energy issues.

### 2.3. FIS (Fuzzy Inference System)

Fuzzy inference systems take inputs and process them based on the predefined rules to produce crisp outputs. Both the inputs and outputs are practical values, whereas the internal processing is based on fuzzy rules and fuzzy operation [32–37].

The processing of the FIS consists of the steps as follows:

1. A fuzzification supports the application of numerous fuzzification methods and converts the crisp input into fuzzy input.
2. A knowledge-based rules and database are formed upon the conversion of crisp input into fuzzy input.
3. The defuzzification fuzzy input is finally converted into crisp output.

The FIS process is shown in Figure 4.

Mamdani-style fuzzy inference requires one to find the centroid of a two-dimensional shape by integrating across a continuously varying function. In general, this process is not computationally efficient. For computational efficiency, a fuzzy singleton is used as a set with a membership function that is unity at a single point on the universe of discourse and zero everywhere else. This Sugeno-style fuzzy inference is very similar to the Mamdani method. However, Sugeno changed only a rule consequently. Instead of a fuzzy set, he used a mathematical function of the input variable [38–47].

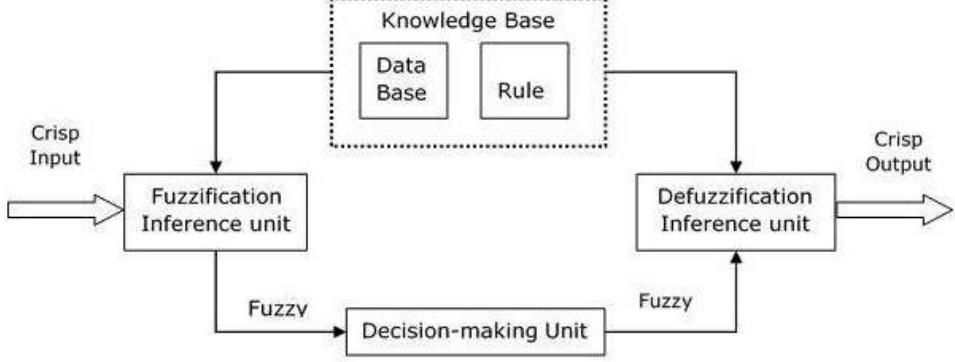

**Figure 4.** A basic FIS (fuzzy inference system) process.

In this paper, we have chosen the Mamdani FIS method because it is widely accepted and is the most commonly used method in FIS, which has the advantage of being intuitive and easy to understand.

## 3. Proposed Protocol: ELEACH-DFL

We propose the ELEACH-DFL which is applied two fuzzy logics for CHs selection and cluster configuration, based on LEACH algorithms, for an energy-efficient network. In the process of selecting CHs, we consider the residual energy and local distance as fuzzy inputs and make rules on which nodes with more residual energy and higher centrality are selected as CHs. After selecting CHs, another FIS is applied to clusters configuration. The remaining energy of Non-CH nodes, distances from CHs, and distances between CHs and BS are used as fuzzy inputs. These two logics allow a network to increase its lifetime.

### 3.1. Cluster Head Selection

When each round starts, each node calculates its own chance value through fuzzy operation. To select the optimal CHs to extend the network lifetime, the remaining energy of each sensor node and the local distance, the sum of the distances from the nodes within a certain range, are regarded as input variables. Figure 5 is the first FIS block diagram for CH selection. The inputs and the output of the first FIS are described in terms of variable names and meanings in Table 2. Membership functions of the FIS are configured like Figure 6 to get optimal performances in terms of energy. The selection of CH candidates is fulfilled using Equation (1), *T(n)* threshold equation of LEACH. Subsequently, after randomly forming a cluster, compare the calculated chance between the CH candidate and the node with the highest chance in a cluster of becoming the CH. The detailed sequence of the CH selection process is shown in below Figure 7.

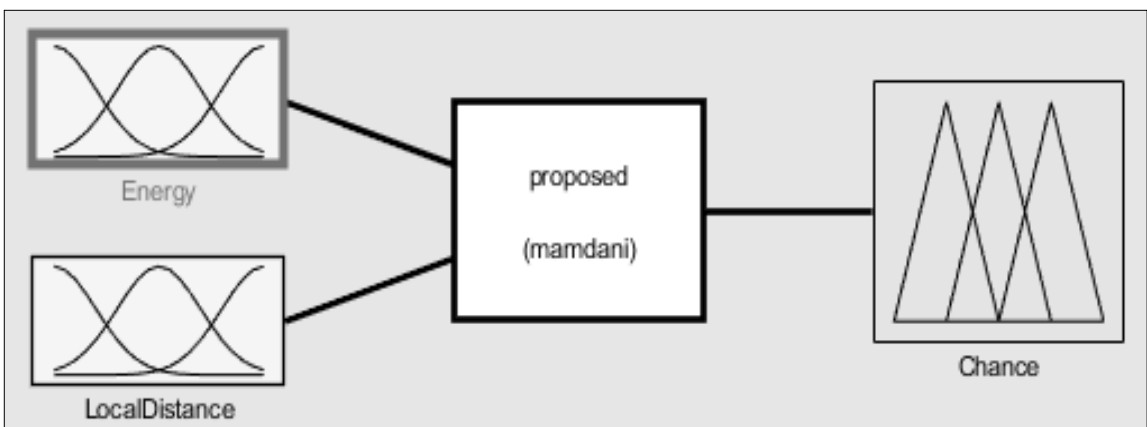

**Figure 5.** First FIS for cluster head (CH) selection.

**Table 2.** Input and output names and meanings.

| Input variables | Energy | Residual energy of the node |
| --- | --- | --- |
| | LocalDistance | The distance to the surrounding nodes within a certain range |
| Output variable | Chance | The threshold value (fuzzy crisp output) to be selected as the CH |

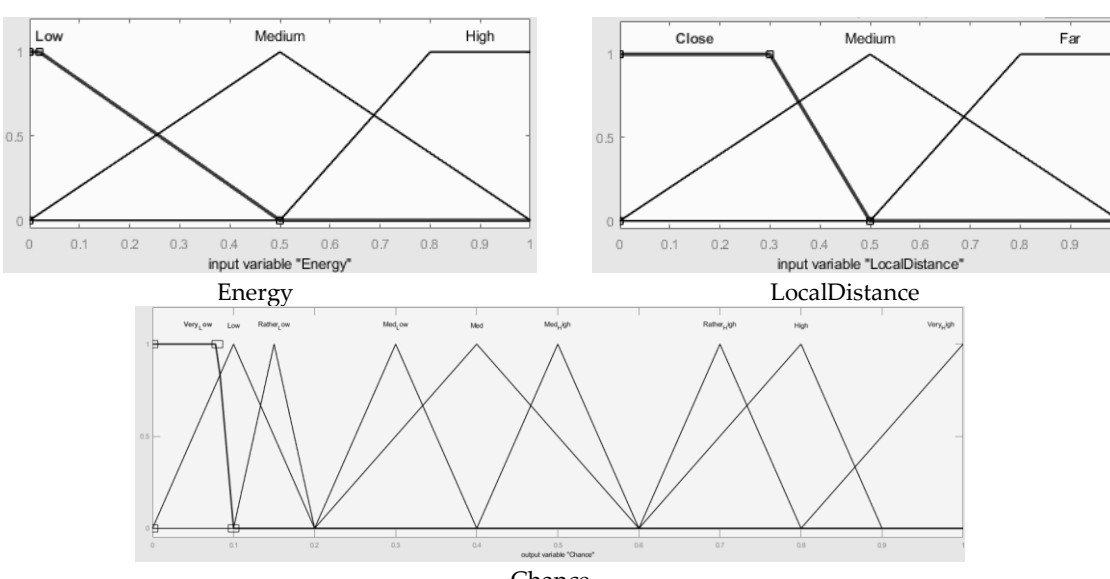

Energy                                    LocalDistance

Chance

**Figure 6.** Membership functions of inputs and output in the first FIS.

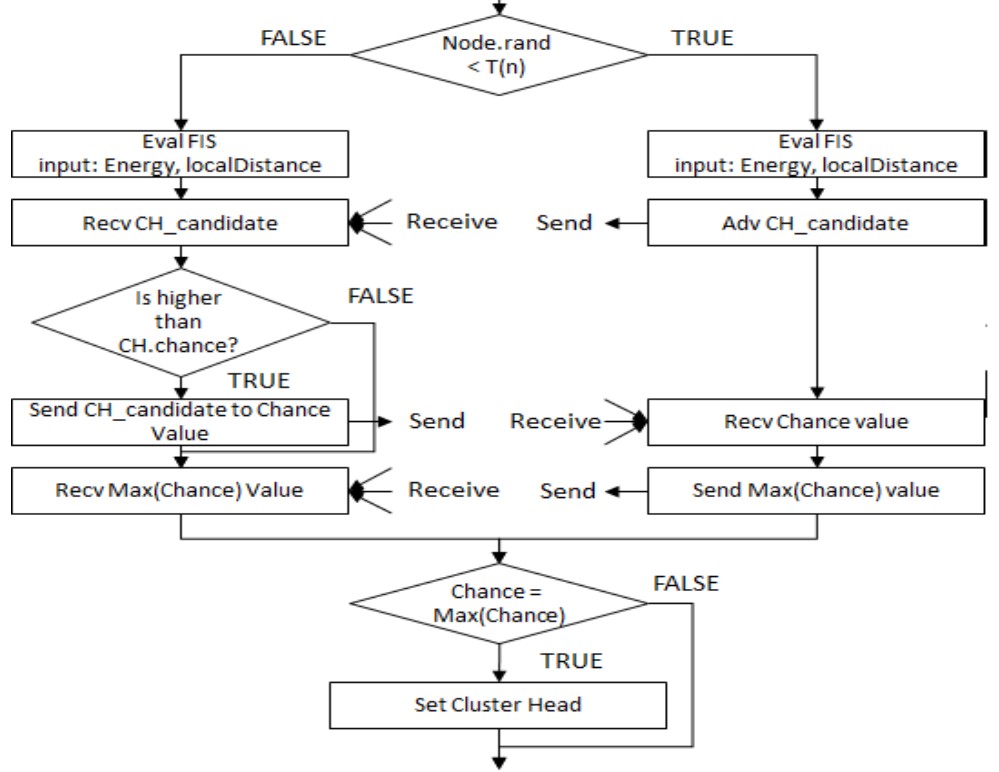

**Figure 7.** Flowchart of the cluster head selection process.

### 3.2. Cluster Configuration

After deciding on CHs, clusters are formed with several non-CH nodes. The second FIS considers the energies of the CHs, distance from BS, and distances between nodes and CHs among on-CH nodes, and obtains the chance values from these three inputs. Figure 8 is the second FIS block diagram for the cluster configuration process. The inputs and output of the second FIS are described in regard to variable names and meanings in Table 3. Membership functions of the FIS are set like Figure 9 to get optimal cluster formations to get optimal performances in terms of energy.

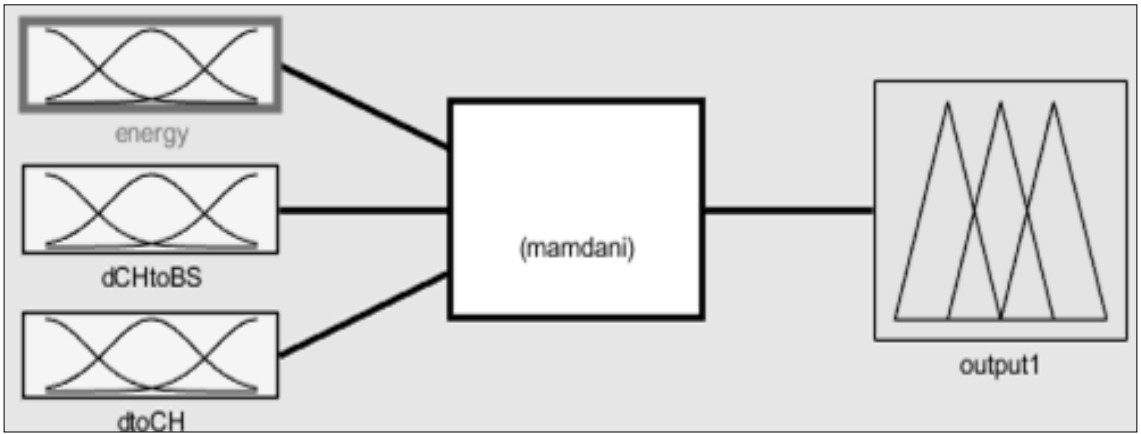

**Figure 8.** Second FIS for cluster configuration.

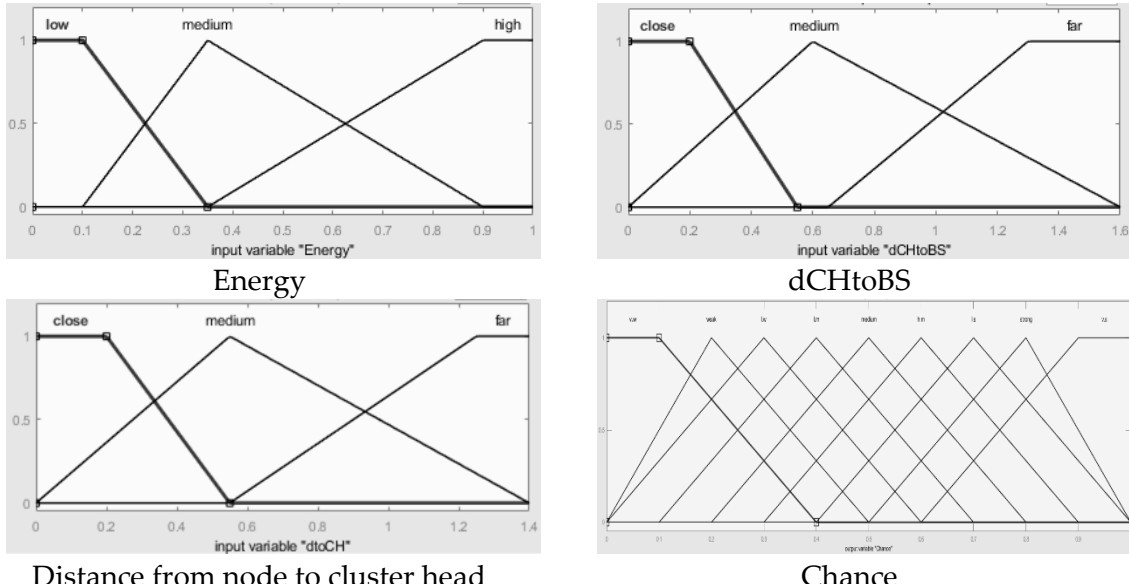

Energy   dCHtoBS

Distance from node to cluster head   Chance

**Figure 9.** Membership functions of inputs and output in second FIS.

Once clusters are formed, each node transmits the measured data to its CH, and then, CHs collect the data and aggregate the collected data if needed or necessary, and send the fuzzed data to the BS. The clustering formation process is described in Figure 10 in detail.

**Table 3.** FIS input and output variables during cluster formation.

| | | |
|---|---|---|
| Input variable | Energy | Residual energy of the node |
| | dtoCH | Distance from the node to the CH |
| | dCHtoBS | Distance from CH to the BS |
| Output Variable | Chance(output1) | The threshold value (fuzzy crisp output) to be selected as the CH |

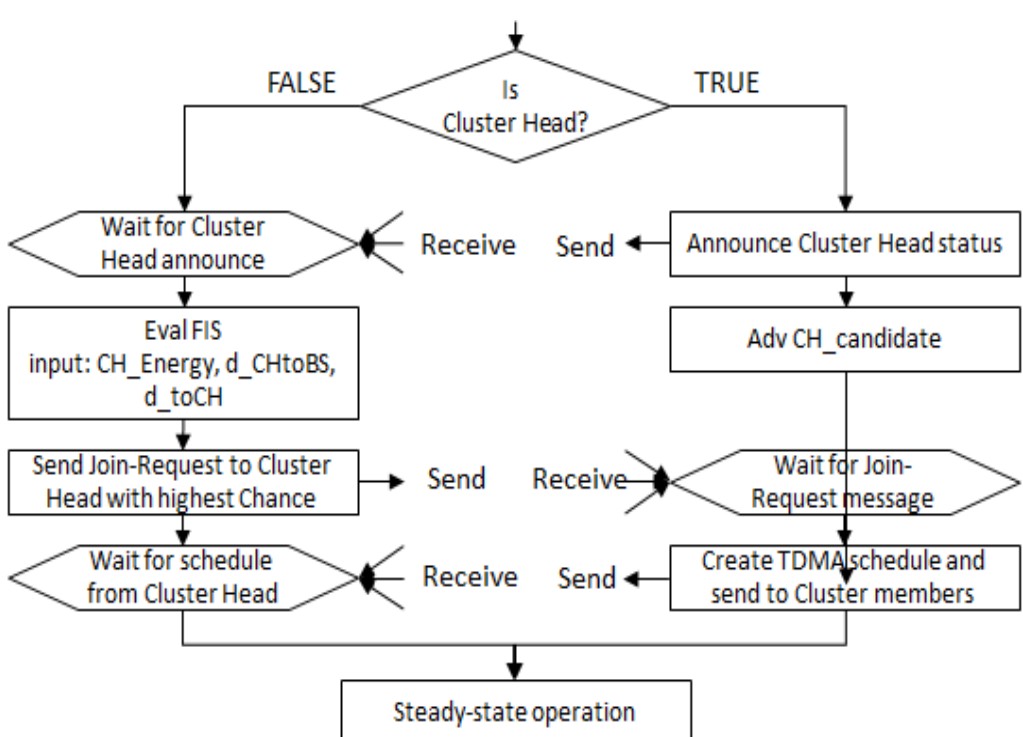

**Figure 10.** Flowchart of cluster configuration process.

The following Figure 11 shows the proposed algorithm's pseudocode. Lines 13–49 of the code are the CH selection and cluster configuration section. Line 15 is the part that determines whether the node's random number is less than the *T(n)* of LEACH if the node is alive. If true, the CH selection section, lines 17–31 will be executed. Line 18 is a FIS operation function, and lines 3–12 are its executing code. If the conditions of line 15 are false, the node becomes a member node and progresses to lines 35–48, which are part of the cluster configuration.

```
1       // Pseudo code
2       // for Every round
3       FIS_CH(Node) // FIS for Cluster Head Election
4       {
5              if Node.isAlive
6                 {
7                      Battery = Node.CurEnergy / Node.InitEnergy
8                      r = sqrt( (Field Size)/(pi * (Total Number of Node) * p) )
9                      Local_Distance = Sum( Node distance in range r )
10                     Node.CHchance = evalFIS.electCH(Battery, Local_Distance)
11                }
12      }
13
14      for i = 1 to (Total Number of Node) Step 1
15      {
16             if Node(i).isAlive and Node(i).rand() < T(n)
17                {
18                     Node(i) is CH.FIS_CH(Node(i))
19                     Adv Cadidate_Msg(Node(i).CHchance)
20                     myCH = Node(i);
21
22                     while ( recv Cadidate_Msg from another Node)
23                     {
24                            if Node(i).CHchance < Node(another).CHchance
25                            {
26                                   myCH = Node(another)
27                            }
28                     }
29
30                     if myCH == Node(i)
31                     {
32                            Adv CH_Message
33                     }
34                }
35             else
36                {
37                     Recv CH_Message
38
39                     if Node.isAlive
40                     {
41                            for j = 1 to (Number of CH) Step 1
42                            {
43                                   Battery = Node(i).CurEnergy / Node(i).InitEnergy
44                                   Distance_BS = Distance Node(RecvCH) to BaseStation
45                                   Distance_CH = Distance Node(i) to Node(RecvCH)
46                                   CHNode(j).chance = evalFIS.formCluster(Battery, Distance_BS,
47                                   Distance_CH)
```

**Figure 11.** *Cont*.

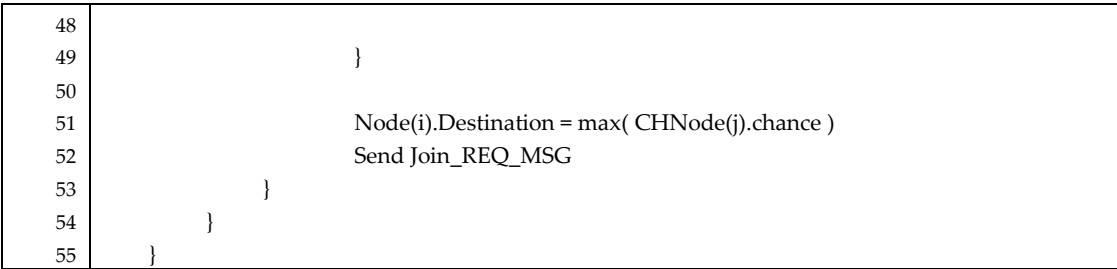

**Figure 11.** Pseudocode of CH and cluster configuration.

## 4. Simulation and Performances

### 4.1. Simulation Conditions

The radio model used in this paper is a two-path propagation model, consisting of two types of reflecting wave and a LOS (line of sight) wave. All sensor nodes were assumed to transmit a certain amount of homogeneous data after converting the measured data into digital signals using an internal A/D converter, and we made a general assumption about the wireless sensor field. That is, the wireless sensor network consists of homogeneous sensor nodes, the distance can be measured according to radio signal strength, and once deployed, the node does not move. Additionally, all sensor nodes have the same initial energy, and the base station is located in the centre or outside of whole the sensor field. Experimental parameters are defined as shown in Table 4 and parameters and meanings for the radio model are summarized in Table 5.

**Table 4.** Experimental parameters.

| Parameter | Value |
| --- | --- |
| Sensor space (M × M) | $100 \times 100$ (m × m) |
| Number of nodes (n) | 100 |
| Initial energy ($E_0$) | 0.5 J |
| Amount of data transferred | 1000 bits |
| Transmission energy ($E_{elec}$) | 50 nJ/bit |
| Data combining energy ($E_{DA}$) | 5 nJ/bit/signal |
| Amplified Energy Factor in LOS ($\in_{fs}$) | 10 pJ/bit/m$^2$ |
| Amplified Energy Factor in multi-path ($\in_{mp}$) | 0.0013 pJ/bit/m$^2$ |

**Table 5.** Parameters and meanings for the radio model.

| Parameter | Meaning |
| --- | --- |
| $l$ | Packet data size |
| $d$ | Transmitting distance between nodes |
| $d_0$ | Limit distance of free space and multi path |
| $d_{toBS}$ | Distance to base station |
| $E_{DA}$ | Energy consumed in data fusion |
| $E_{elec}$ | Factors related to digital coding, modulation, filtering, signal spreading, etc. |
| $\in_{fs}$ , $\in_{mp}$ | Energy required for free space or multi-path amplification |
| $d_{toCH}$ | Distance to cluster head |

We used MATLAB to simulate the proposed ELEACH-DFL and compare it with the LEACH protocol in terms of energy consumption. We evaluate and compare FND (first node dead), HND (half node dead) and LND (last node dead) values, which are normally used to evaluate energy performance.

### 4.2. Performance Comparison

Table 6 shows the constellation of sensor nodes and clusters when the proposed protocol is adopted, when the location of BS is set to (50, 50) at the inside of the sensor field at first, second,

and FND rounds. The FND takes place in 3942 rounds. The FND of the LEACH protocol is 2983. The FND efficiency improvement rate of ELEACH-DFL over the LEACH protocol is approximately 32%. The results are confirmed in Figures 12 and 13.

**Table 6.** Network lifetime comparisons between protocols.

| Round | FND | 80% Alive | 50% Alive |
|---|---|---|---|
| LEACH | 2983 | 3322 | 3740 |
| ELEACH-DFL | 3942 | 4320 | 4439 |
| Improvements | 32% ▲ | 30% ▲ | 19% ▲ |

**Figure 12.** Constellation of nodes and clusters' configuration.

Table 7 shows the constellation of sensor nodes and clusters when the proposed protocol is adopted, when the location of BS is set to (50, 150) at the outside of the sensor field at first, second, and FND rounds.

**Table 7.** Network lifetime comparisons between protocols.

| Round | FND | 80% Alive | 50% Alive |
|---|---|---|---|
| LEACH | 1412 | 3322 | 2654 |
| ELEACH-DFL | 3654 | 4320 | 4012 |
| Improvements | 159% ▲ | 30% ▲ | 51% ▲ |

The FND takes place in 3654 rounds in the proposed algorithm. The FND of the LEACH protocol is 1412. The FND improvement rate of ELEACH-DFL over LEACH is approximately 159%. When compared to around 80 percent of nodes being alive and HND, the proposed algorithm is improved by 30% and 51% respectively. The results are confirmed Figures 14 and 15.

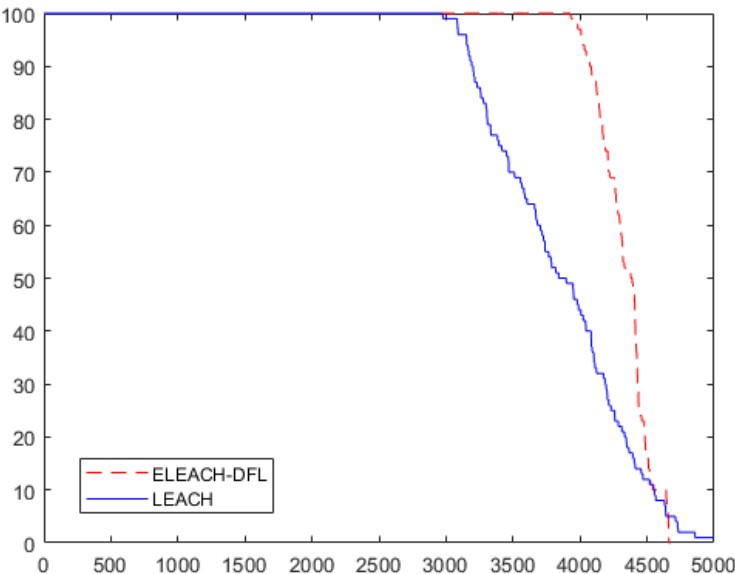

**Figure 13.** Network lifetime comparison between protocols (base station (BS)): inside the sensor field).

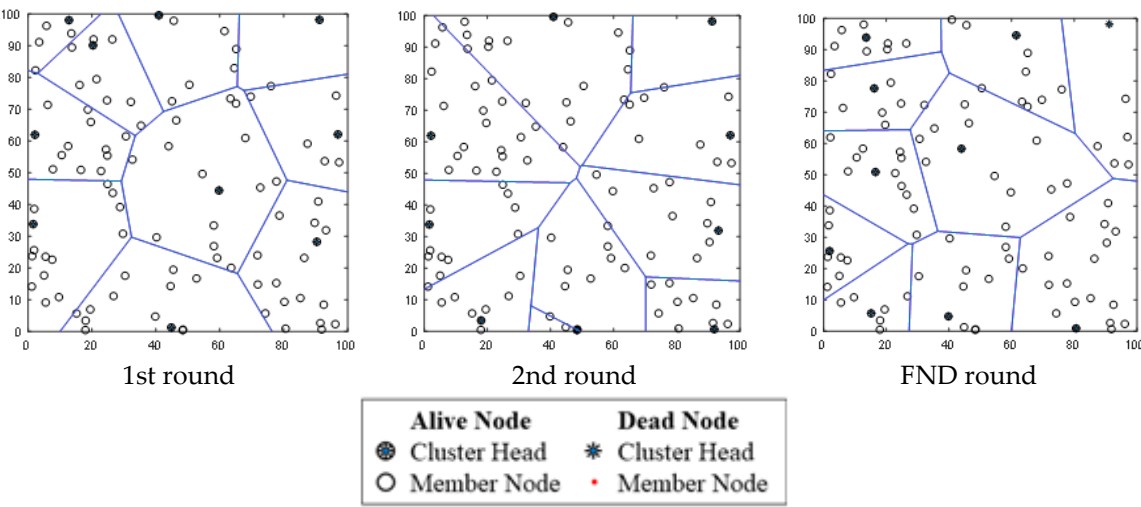

**Figure 14.** Constellation of nodes and clusters' configuration.

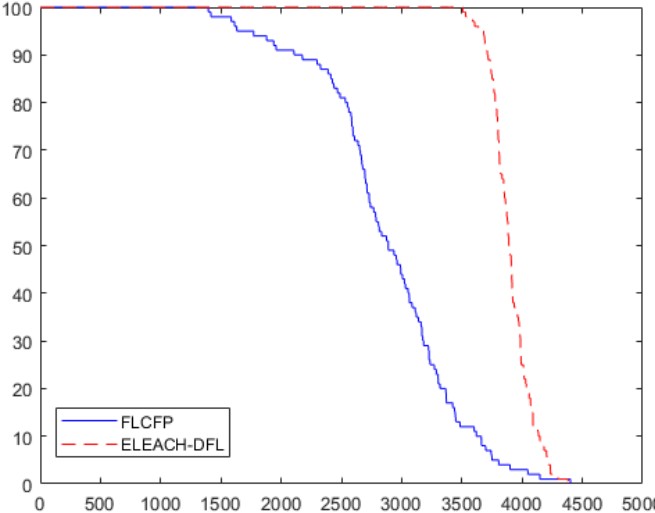

**Figure 15.** Network lifetime comparison between protocols (BS: outside the sensor field).

The above experiments did not take into account the calculation quantity or the reliability of transmission; however, as the results indicate, the proposed algorithm has better energy efficiency characteristics than conventional LEACH regardless of the location of the BS.

## 5. Conclusions

For wired sensor networks, wiring sensors to the user may cause problems and inconvenience regarding mobility. This led to the use of wireless sensor networks to proceed with body sensor networks, which should take into account the problem of power efficiency. Specifically, in body sensor networks, using implanted sensors and optimizing the energy consumption can keep sensors alive long.

The LEACH protocol is energy efficient before the first node is dead. However, there is a sharp drop in energy efficiency after FND. The proposed protocol, ELEACH-DFL, has the advantage of maintaining energy efficiency even after FND occurs, while considering cluster configuration and CH selection separately. In a cluster-based routing WSN protocol, network lifetime is severely affected by the configuration of clusters and the location of CHs. Without considering these, LEACH improved only the problem of one node being selected continuously as the CH by ensuring that all nodes are selected evenly.

Therefore, ELEACH-DFL proposes using a fuzzy logic to improve CH selection issues. Thus, ELEACH-DFL allows the optimal CH to be selected by considering the energy of each node and the location or density of the nodes. The ELEACH-DFL (extended LEACH-dual fuzzy logic) proposes both CH selection and cluster configuration methods. When selecting a CH, the CH candidate was firstly selected using the threshold equation, and the node with a highest chance was determined as the CH by comparing the remaining energies of the nodes among the candidates and the distances of the near nodes together. After CH selection, it was decided that when each non-CH candidate node participates in a cluster, it should participate in the appropriate cluster based on the remaining energy of the CH, distance from BS, and distance to the CH.

We compared the proposed ELEACH-DFL with LEACH in terms of energy efficiency, FND, and HND. In a field size of $50 \times 50$, the FND efficiency improvement rate of ELEACH-DFL versus LEACH protocol is approximately 32%. In addition, in a field of $50 \times 150$, the FND efficiency improvement rate of ELEACH-DFL versus the LEACH protocol is approximately 159%. The proposed algorithm has better energy efficiency characteristics than conventional LEACH, regardless of the location of the BS. It would be better for further research to consider the computational amounts and reliability of data transmission simultaneously.

**Author Contributions:** The authors of D.Y.Y. and T.K. conducted a basic survey on the research, and the overall study of ELEACH-DFL was led by the authors of J.-Y.L. and D.L., and the thesis research was conducted overall by the author of S.Y.P. All authors have read and agreed to the published version of the manuscript.

**Funding:** No external funding available.

**Conflicts of Interest:** The authors have no conflicts of interest.

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
