# Peer review of "An Energy Efficient Enhanced Dual-Fuzzy Logic Routing Protocol for Monitoring Activities of the Elderly Using Body Sensor Networks"

_electronics, doi:10.3390/electronics9050723_

Round 1
Reviewer 1 Report
Comments for the Authors (continue on another sheet, if necessary):
- Computational amount, delay and reliability should be considered in future works.
- Increase the reference literature a little more on the II. Related Researches : LEACH and Fuzzy logic
- Correct the blurred parts of the pictures and formulas in the paper. 
 Figure 2 : hard to read...enlarge the font size.small , and figure 3 or amend the blurred parts of the Figures.
 Example : Figure 2. Example of patient monitoring in a Wireless Body Area Network
 Figure 3. One Round of Flow-chart of LEACH
- Correct the entire sentence and paragraph regularly within the paper.
- Organize the Shudo code and re-set it by aligning paragraphs.
- Table 2/3 : add line for separation between input and output variables.
Author Response
| 
 Requests  | 
 Responses  | 
| 
 1. Computational amount, delay and reliability should be considered in future works.  | 
 In future study : It would be better for further researches to consider the computational amounts and reliability of data transmission simultaneously.  | 
| 
 2. Increase the reference literature a little more on the II. Related Researches : LEACH and Fuzzy logic  | 
 Added : References – 22-25, 28-31  | 
| 
 3.     Correct the blurred parts of the pictures and formulas in the paper.   | 
 Modified : It has been corrected clearly and accurately.  | 
| 
 4. Correct the entire sentence and paragraph regularly within the paper.  | 
 Checked again : It has been corrected clearly and accurately.  | 
| 
 5. Organize the Shudo code and re-set it by aligning paragraphs.  | 
 Re-organized : Figure 11.  | 
| 
 6. Table 2/3: add line for separation between input and output variables.  | 
 Lines added for separation  | 
Reviewer 2 Report
- What problem in clustering is not mentioned in the abstract.
 - Latest references should be included and the comparability of the methods described in 2.1 with the method mentioned in this article is not obvious.
 - Comparison with the recent work is missing in section 4.
 - In table 3, ‘The threshold value to be selected as the cluster head’ should be expressed more accurately.
 - Whether Table 6 and 8are described as Figure 6 and 8 is more appropriate?
 - The introduction of fuzzy systems in this paper is too simple. Mamdani fuzzy system is mentioned in Figures 5 and 7 but not described in detail.
 - The article is insufficient in theory and novelty.
 - There are some mistakes in this paper, such as ’d blood glucose’.

Round 2
Reviewer 2 Report
1 Typography is messy and the position of caption is incorrect.
2 Methods described in 2.1 are not for clustering techniques.
3 A lot of improved algorithms for clustering of leach protocol can be compared with the method proposed in this paper.
4 Can the fuzzy system be further described by formula or code?
Author Response
1. What problem in clustering is not mentioned in the abstract?
-> Added clustering technique in abstract : Considering the importance of sensor nodes in a specific environment for improving the network's lifetime, the protocol based on LEACH (Low Energy Adaptive Clustering Hierarchy) algorithm would be proposed. Because avoiding long distance transmission, clustering technique is an efficient algorithm for prolonging the lifetime of sensor networks. Therefore, this paper suggests an Enhanced LEACH-Dual Fuzzy Logic (ELEACH-DFL) protocol based-on clustering for CH (Cluster Head) selection and cluster configuration in wireless sensor networks.
2. Latest references should be included and the comparability of the methods
-> Added in section 2, References – 22-25, 28-31
3. described in 2.1 with the method mentioned in this article is not obvious.
-> The ELEACH-DFL protocol was proposed to indicate energy efficiency.
4. Comparison with the recent work is missing in section 4.
-> We compared LEACH with the proposed algorithm. LEACH is a representative algorithm in clustering method
5. In table 3, ‘The threshold value to be selected as the cluster head’ should be expressed more accurately.
->
Expressed : The threshold value (fuzzy crisp output) to be selected as the CH
6. Whether Table 6 and 8are described as Figure 6 and 8 is more appropriate?
-> Changed : Figure 6. Membership Functions of Inputs and Output in 1st FIS
And Figure 9. Membership Functions of Inputs and Output in 2nd FIS
7. The introduction of fuzzy systems in this paper is too simple. Mamdani fuzzy system is mentioned in Figures 5 and 7 but not described in detail.
-> Modified and added details
8. The article is insufficient in theory and novelty.
-> This paper proposes a cluster-based routing protocol which is suitable to WBAN while analyzing energy efficiency issue in data transmission. The simulation and analysis results address that the enhanced algorithm reduces the energy consumption effectively and extends the lifespan of the entire network. The ELEACH-DFL protocol was proposed to indicate energy efficiency.
9. There are some mistakes in this paper, such as ’d blood glucose’.
-> Corrected : For example, the elderly's vital sign, such as, ECG (Electrocardiogram), blood pressure, heart rate, and blood glucose, can be used as a measure of their well-being and are all critically important for remote elderly care in tracking their physical and cognitive capabilities.

Round 3
Reviewer 2 Report
1 Typography is messy and the position of caption is incorrect.
2 Methods described in 2.1 are not for clustering techniques.
3 A lot of improved algorithms for clustering of leach protocol can be compared with the method proposed in this paper.
4 Can the fuzzy system be further described by formula or code?
5 You should revise it according to the second opinion, but reply to the first .
Author Response
| 
 Requests  | 
 responses  | 
| 
 1. Typography is messy and the position of caption is incorrect.  | 
 Several figures and tables have been placed in correct. Ex. Table 2,3,4,5 / Figure 12,14  | 
| 
 2. Methods described in 2.1 are not for clustering techniques.  | 
 “AnyBody” is a data gathering protocol that uses clustering to reduce the number of direct transmissions to the BS. Each node sends out a hello message in which it puts its unique identifier. No message is sent if the node has a higher density than all its neighbours. The join messages are relayed by the receiving nodes until it reaches a node with highest local density. Through this messaging process, the CH is elected with highest density and nodes are grouped into clusters. After clustering, nodes need to be interconnected. All these connections between CHs will form a virtual backbone network. Finally, the routing paths are set up. Network in this protocol has a constant number of clusters even though the number of sensor nodes are increased. Line : 206~213  | 
| 
 3. A lot of improved algorithms for clustering of leach protocol can be compared with the method proposed in this paper.  | 
 Yes, We will try to compare performance with various clustering protocols in the future study.  | 
| 
 4. Can the fuzzy system be further described by formula or code?  | 
 The fuzzy theory was applied in two main points. One implemented fuzzy theory when selecting cluster heads. Inference (Figure.5) was made in consideration of Energy and Local Distance, and as a result of the purge, the Chance value of Figure 6 was obtained. There is also Table 2, which organizes this. Secondly, we implemented fuzzy theory in cluster configuration. Inference (Figure.8) was made in consideration of Energy, dCHtoBS, and dtoCH. As a result of spreading, the Chance value of Figure 8 was obtained, and Table 3 is arranged.  |